# Metal–Metal Bond in the Light of Pauling’s Rules

**DOI:** 10.3390/molecules26020304

**Published:** 2021-01-08

**Authors:** Elena Levi, Doron Aurbach, Carlo Gatti

**Affiliations:** 1Department of Chemistry, Bar-Ilan University, Ramat-Gan 5290002, Israel; doron.aurbach@biu.ac.il; 2CNR-SCITEC Istituto di Scienze e Tecnologie Chimiche “Giulio Natta”, sezione di via Golgi, via Golgi 19, I-20133 Milano, Italy; 3Istituto Lombardo Accademia di Scienze e Lettere, via Brera 28, I-20121 Milano, Italy

**Keywords:** bond order, metal–metal bond, bond valence sum, steric effect, electrostatic effect, material stability

## Abstract

About 70 years ago, in the framework of his theory of chemical bonding, Pauling proposed an empirical correlation between the bond valences (or effective bond orders (BOs)) and the bond lengths. Till now, this simple correlation, basic in the bond valence model (BVM), is widely used in crystal chemistry, but it was considered irrelevant for metal–metal bonds. An extensive analysis of the quantum chemistry data computed in the last years confirms very well the validity of Pauling’s correlation for both localized and delocalized interactions. This paper briefly summarizes advances in the application of the BVM for compounds with TM–TM bonds (TM = transition metal) and provides further convincing examples. In particular, the BVM model allows for very simple but precise calculations of the effective BOs of the TM–TM interactions. Based on the comparison between formal and effective BOs, we can easily describe steric and electrostatic effects. A possible influence of these effects on materials stability is discussed.

## 1. Introduction

In 1929, based on the first structural studies, Pauling formulated five simple rules to rationalize chemical bonding in minerals [1]. One of them is a bond order (BO) conservation principle or a rule of local electroneutrality, which states that the cation charge in the crystal structure should be compensated by the negative charge of adjacent anions and vice versa. Consequently, the BO sum, Σ *BO_i__j_*, or the bond valence sum, BVS = Σ *s_i__j_*, (sum of the bond strengths according to Pauling) around atom *i* should be equal to its valence, *V_i_*:Σ *BO**_ij_*** = Σ *s_ij_* = *V_i_*(1)

According to Equation (1), the BO can be defined as a number of valence electrons of an *i* atom associated with a given bond. In 1947 Pauling related the *BO_i__j_* to respective interatomic distance, *R_ij_*, by empirical constants [2]. This correlation can be written as follows:*s_ij_* = exp [(*R*_0 *ij*_ − *R_ij_*)/*b_ij_*],(2)

The constants *R*_0_
*_ij_* and *b_ij_* are known as bond valence (BV) parameters, transferable for a given atom pair in different compounds. They represent an effective repulsion between *i* and *j* atoms and the softness of the bond, respectively [3,4].

Equations (1) and (2) are widely used in crystal chemistry in the framework of the bond valence model (BVM) [3,4], mainly in order to verify the crystal structure solution or to determine the oxidation state of atoms. The BV parameters for a large combination of the atom pairs were proposed [5]. For compounds with localized bonds, it was shown that, in good accordance with Equations (1) and (2), the BVSs calculated with these BV parameters are close to the expected number of valence electrons (Although some studies mark the limited predictive power of the Pauling Rules [6]). In contrast, the Pauling rules and the BVM were for a long time deemed as inapplicable to compounds with delocalized bonds, e.g., metal cluster compounds (i.e., materials with metal–metal, TM–TM (TM = transition metal), interactions) [4,7]. Only recently, the efficacy of the model for such compounds was shown [8,9,10,11,12,13,14,15,16,17].

This paper briefly summarizes the advance in the application of the BVM to compounds with metal–metal bonding, also presenting novel examples and focusing on the validity of Pauling’s rules for these materials. It is shown herein how structural and bonding peculiarities of these materials can be described and understood based on Pauling’s approach.

## 2. Methods: Determination of the BV Parameters for the Metal–Metal Bonds

As is clear from Equation (2), the accuracy of the BVS values depends mostly on the choice of BV parameters. A few attempts to use quantum chemistry considerations for Pauling’s rules and calculation of the BV parameters are known [18,19,20,21,22], but till now, the BVM remains empirical. Different sets of the BV parameters for the cation-anion pairs were proposed [5], but all of them are based on the direct use of Equations (1) and (2). It means that the BV parameters for a given atom pair are chosen to satisfy a local electroneutrality in a large number of compounds with this pair. As discussed below, such an approach is convenient only for solids without steric constraints. Since metal–metal bonds are commonly too short of matching their atomic surrounding in the crystal structure; it is difficult to find the TM_n_ cluster compounds without lattice strains. In this case, the direct method cannot be used, but Equations (1) and (2) can be applied indirectly based on the valence compensation (see Equation (3) below). In this case, the BVS of the TM–TM bonds, Σ *s _TM–TM_*, is accepted to be equal to the difference between the total number of the TM electrons responsible for bonding, *l_TM_*, (e.g., six valence electrons for Mo atom in Mo_6_-cluster compounds) and the BVS of the TM–L bonds, Σ *s_TM-L_*. The latter can be calculated with conventional BV parameters [23].

For the determination of the BV parameters, we also used the large sets of quantum chemistry data on the effective BOs for the TM–TM bonds, which are available in the literature. In this case, the BV parameters were fitted based on the BO exponential trend vs. respective interatomic distances. The approximate values of *R*_0_
*_ij_* for the TM pairs were also calculated using empirical constants proposed by O’Keeffe and Brese in 1991 for most of the chemical elements (*b_ij_* of 0.37 Å is assumed to be the same for all the *i* and *j* pairs) [24]. It was shown that all three methods (based on BO conservation, on quantum chemistry data and O’Keeffe’s constants) result in the close values of the BV parameters [16], but the first approach seems the most reliable for the moment. In particular, the evaluation of BV parameters from quantum chemical data could largely benefit from the (hopefully future) availability of a large set of BO data computed using wavefunctions of comparable quality and a single BO recipe. However, it should also be mentioned that steric and electrostatic effects in cluster compounds are commonly so pronounced that the choice of the BV parameters is less crucial as compared to the case of localized bonds.

Details of the electron counting for various cluster compounds can be found in our previous works (see, for example, Ref. [14]).

## 3. Results and Discussion

### 3.1. Validity of Equation (2) for Localized and Delocalized Bonds

To validate the BVM application to the delocalized bonds, the first question that should be discussed is “What was the reason to reject Pauling’s rules for compounds with metal–metal interactions?”. Cotton, in the book “*Multiple bonds between metal atoms,*” wrote [7]: “It is a general qualitative rule in chemistry that bond lengths and bond orders are inversely related… However, there is no a priori reason to expect that similar procedures will (or will not!) work in the very different realm of metal-to-metal bonds. Experience is the only test, and experience thus far has shown that M–M bonds cannot usefully be treated in such a way. We condemn as foolish and hopeless any effort to associate a unique, quantitative bond order with each and every metal–metal internuclear distance.” As an argument for this hard statement, Cotton showed that the same formal BO of the metal–metal bond might be associated with effectively different bond lengths. Indeed, e.g., the length of the Cr–Cr quaternary bond (formal BO = 4) in Cr_2_ complexes ranges from 1.830 to 2.541 Å [25].

The problematic point in Cotton’s argument is his correlation of the bond lengths to the formal BOs, while Pauling’s Equation (2) relates them to the effective BOs. (The formal BOs for the cation–anion bonds are those expected from the formal oxidation state of the ions. For the metal–metal bonds, the formal BOs should be evident from the structure of molecular orbitals. It is defined as half of the difference between the numbers of bonding and antibonding electrons.) In contrast to the localized bonds, the effective BOs of the metal–metal interactions can significantly differ from the formal values. As it was shown [26], the difference may reach 1 valence unit (v.u.). As an example, Figure 1 presents formal and effective BOs of the Cr–Cr bonds as a function of their lengths. The effective BOs (Mayer, Wiberg, delocalization indices, etc.) were calculated by different authors for various compounds (the references of these studies can be found in [14]). As can be seen, in contrast to the formal BOs, the calculated BOs clearly follow exponential decay, in spite of the high dispersion of the data. A similar analysis of the effective BOs for various atomic pairs obviously confirmed the validity of Equation (2) for both localized and delocalized bonds [13]. The BV parameters of the TM–TM bonds obtained by fitting the exponential curves allow for a simple but careful calculation of the effective BOs from the TM–TM distances by Equation (2) [14,15,16].

### 3.2. Validity of Equation (1) for Compounds with Metal–Metal Bonds

#### 3.2.1. Stretching of Metal–Metal Bonds Evident by Valence Violations

In 1992 Brown proposed important comments to the rationalization of Equation (1), showing that this equation is valid only for the unstressed bonds [27]. Lattice strains caused by the steric mismatch between different atoms in the crystal structure result in the deviation of the BVS Σ *s_i__j_*, from the expected *V_i_* value. Thus, the difference Σ *s_i__j_* − *V_i_* may serve as a measure of the lattice strain and material instability [3,4,27]. A great advantage of this method is that, in combination with structural analysis, it shows the source of material instability. The problem is that, for the compounds with localized bonds, the Σ *s_i__j_* − *V_i_* difference is commonly small. As a result, in practice, this method needs a very careful choice of the BV parameters, as well as experimental interatomic distance, *R_ij_*, otherwise the Σ *s_i__j_* − *V_i_* difference may be caused by inaccuracy or better indeterminacy in the BVS calculations. In spite of this difficulty, the method was successfully used for perovskites [3,4,27] and electrode materials [28] to explain or even to predict their instability.

The existence of stresses in the metal–metal bonds embedded in the inorganic atomic framework was firstly marked by Schäfer and Schnering [29]. Corbett defined a steric mismatch between short metal–metal bonds and closed-shell anion surrounding as “matrix effect” and tried to use Equation (2) to describe it [30,31]. It is interesting to compare the BV parameters proposed by Corbett based on the respective distances in metals with those calculated by Levi et al. using the conservation principle, as well as numerous quantum chemistry data of effective BOs available in the literature [14] (Table 1). As can be seen, two sets of the BV parameters are relatively close to each other, but the *R*_0_ values proposed by Corbett are systematically bigger, resulting in the underestimation of the stretching in the metal–metal bonds.

Using the BVM analysis and structural approach, we can explain the bonding peculiarities of cluster compounds, e.g., the expansion of the TM_n_ clusters with the size of surrounding anions. At first glance, such expansion is strange because the cluster size should depend only on the number of bonding (and antibonding) electrons, but not on the anions’ size. However, from structural considerations, it is clear that the higher the ligand size, the larger is the void formed by anions, and the higher is the stretching of the TM–TM bonds in the TM_n_ cluster that occupies this void. To illustrate this statement, Figure 2a compares the BVS and the formal number of valence electrons for the Mo–Mo bonds in isostructural compounds, Cu_2_Mo_6_L_14_ (L = Cl, Br and I). A structure of the cluster unit of this compound is presented in the upper inset of Figure 3. An increase in the ionic radius from Cl (1.81 Å) to Br (1.96 Å) and to I (2.20 Å) results in longer Mo–Mo bonds in the Mo_6_ cluster, and, respectively, in lower BVS of these bonds, while the formal valence remains the same. Thus, a general conclusion can be drawn: the stretching of the metal–metal bonds is responsible for the unusually high difference between their formal and effective BOs (Figure 1 and Figure 2). The higher is the stretching; the lower are the effective metal–metal BOs (and respective BVSs).

#### 3.2.2. Compensation of the Valence Violations for the Bonds around Transition Metals

A reasonable question arises: “Does the stretching of the metal–metal bonds decrease the material stability?” The answer is rather negative because the valence deficiency caused by stretching is compensated by the valence excess related to the compression of the metal–ligand bonds. As can be seen from Figure 2b, the effective BVSs for the Mo-L bonds in Cu_2_Mo_6_L_14_ is effectively larger than the formal ones, and their rise with the ionic radius of the ligands corresponds to the BVS drop for the Mo–Mo bonds. The total BVS of the Mo atoms for all three compounds are close to 6, i.e., to the number of valence electrons of the Mo atom (Note that for Mo_6_ clusters in these compounds, all the electrons are bonding). In the general case, in accordance with the BO conservation principle, the total BVS for the metal atom, which includes the metal–metal (TM–TM) and metal–ligand (TM–L) bonds, is close to the expected number of valence electrons, *l_TM_*, responsible of bonding:*BVS_TM_* = Σ *s_TM-TM_* + Σ *s_TM-L_* = *l_TM_*.(3)

Moreover, it was shown that the matrix effect is commonly associated with a more symmetric distribution of valence electrons and electron density around transition metals [10,14]. For example, in the compound Cu_2_Mo_6_I_14_ with an octahedral Mo_6_ cluster, the Mo atom is bonded to four Mo atoms and five I atoms (see the upper inset in Figure 3). The formal valence distribution is four electrons for the Mo–Mo bonds and two electrons for the Mo-I bonds. Taking into account similar lengths of all the bonds of the same type, the formal BOs should be close to 1 v.u. for the Mo–Mo bond and 0.4 v.u. for the Mo-I bond. Due to the bond strains, in real Cu_2_Mo_6_I_14,_ the bond valences are 0.62 and 0.68 v.u., respectively. It is logical to suggest that such symmetric valence distribution caused by the matrix effect impacts the stability of the cluster units. Indeed, it is known that, in spite of the strained bonds, cluster core, Mo_6_L^i^_8_, remains stable even after the dissolution of cluster compounds in different solvents [32].

#### 3.2.3. Clusters as Single Cations with Nonuniform BVS Distribution on Their Ligands

The most unusual result that follows from the BVS calculations for compounds with metal–metal bonds is that, in contrast to the TM atoms, Equation (1) is not valid for the BVS of the separate ligands, but only for a given cluster unit as a whole. Moreover, the BVS distribution on the ligands in many cluster compounds is extremely nonuniform, with a high deviation of the BVS from the expected values. As an example, Figure 3 presents cluster contribution to the valence of the ligands, *BVS_L_* = Σ *s_TM-L_*, as a function of their distance from the cluster center for the Re_6_-chalcogenides. (Here, *BVS_L_* is related only to the bonds between the ligand and a separate cluster.) As can be seen, this contribution to the inner ligands is about three times higher than that for the outer ligands. This result is not surprising, taking into account the structure of the cluster units: close values of the TM–L distances for the inner and outer ligands and different coordination around chalcogen atoms: three Re atoms for the inner ligands and only one for the outer ligands (see the upper inset in Figure 3). It means that three Re^3+^ cations contribute their valence electrons in the bonding with the inner ligand, but only one Re^3+^ cation assists in the bonding with the outer ligand.

It is interesting to note that the BVS distribution on the ligands can be easily understood if, in our bonding scheme, we will replace the TM_n_ cluster with a single cation located in the cluster center (see the low inset in Figure 3). It can be shown that the curves in Figure 3 follow Equation (2), where *R_ij_* is the bond length between the imaginary TM_n_-cation and its ligand, and *R_0__ij_* and *b_ij_* are new BV parameters. These parameters calculated for a set of the TM_6_-cluster compounds are presented in Table 2. A large value of *b_ij_* (more than 1 Å in our case) is commonly assigned to a high difference in the electronegativity of the *i* and *j* atoms [33]. According to the *R*_0_ values, the sizes of the imaginary TM_n_ cations are comparable with the size of such cations like Cs^+^ or Ba^2+^, but the formal charge (i.e., Σ *BO_i__j_* = Σ *s_i__j_* = *V_i_*, Equation (1)) is much higher, ranging commonly from 11 to 17 for the Nb_6_, 12–16 for Mo_6_, 12–22 for W_6_ and being equal to 18 for Re_6_. Due to the large ionic charge, the polarizing power of the clusters should be high. Consequently, the inner ligands located closer to the cluster center should have an effectively larger charge than the outer ones.

To avoid any potential misunderstanding, it is worth recalling that the BVM notion of ionic charge does not bear any direct relation with that of atomic charge customarily adopted in theoretical and computational chemistry. The former, derived from the BVS, measures the actual “valence” of an atom as opposed to its formal value, whereas the latter is simply given by the difference between the atom’s nuclear charge and its fractional number of electrons, evaluated with some suitable recipe (Mulliken’s, Bader’s charges, etc.).

#### 3.2.4. Charge Transfer from the Cluster to the Ligands and between the Ligands

In the previous section, we dealt with formal charges of the clusters, but, based on the anion BVSs, we can calculate their effective charges and compare them to the formal ones. For example, the oxidation state of Re in the Re_6_-cluster compounds is +3, and the formal charge of the Re_6_-cation is equal to +18, while the effective charge is +26.4 for sulfides, +27.8 for selenides and +31.1 for tellurides (average data for few compounds). For the Mo_6_-cluster compounds with the Mo oxidation state of +2 (the formal charge of +12), the effective charge is equal to +18.2 for chlorides, +18.5 for bromides and +20.0 for iodides. Thus, the electron redistribution around TM metals in the cluster units, associated with the matrix effect, results in the charge transfer from the cluster (and from the outer ligands) to the inner ligands. As expected from the matrix effect, the value of the charge transfer from the cluster depends on the size of the inner ligands (Figure 4a), but we can also relate it to the ligand electronegativity (Pauling scale) (Figure 4b). The higher the latter, the smaller is the charge transfer. The influence of the outer ligands on the charge transfer is much less pronounced (Figure 5). Moreover, the cluster contribution to the BVSs of the outer ligands is almost unaffected by their composition, allowing easy mutual substitution of these ligands in the synthesis.

Again, it is important to stress that analogously to the notion of “ionic charge”, that of “charge transfer” has in the BVM context a completely different meaning relative to the same term assumed in theoretical/computational chemistry. BVM’s charge transfer is related to an increase of the BVS of subsets of bonding interactions (in the present case, the TM–L^i^) at the expense of the other ones in a system, while in theoretical/computational chemistry, the term “charge transfer” simply expresses the (fractional) number of electrons transferred from one to another system’s moiety, as a result of the chemical interaction among these, originally isolated, moieties. As a consequence, just opposite to the BVM case, the higher the ligand electronegativity, the larger is the charge transfer from the TM cluster to the ligands in the computational/theoretical chemistry language.

#### 3.2.5. Ligand Valence Violations as a Source of Material Instability

The difference in cations’ contribution to the valence of anions is a normal phenomenon for solids. For example, in orthorhombic NaMnO_2,_ two crystallographically different oxygen atoms have different bonding to the Mn^3+^ cations, as well as different Mn input to the oxygen valence: about 0.7 and 1.1 v.u. However, this difference is compensated by the respective contribution of Na cations in the oxygen BVS. Otherwise, according to Pauling, the material should be unstable. The BV analysis of cluster compounds shows that it is much more difficult to compensate for the difference between the effective anion charge (BVS) and the formal one, especially for large clusters. Various mechanisms of such compensation for compounds with octahedral clusters are described in refs. [8]. They include (i) formation of the mixed chalcogen-halogen compositions, with the preferential occupation of the inner and outer sites in the cluster units by chalcogen and halogen atoms, respectively; (ii) occupation of the outer sites by chalcogen atoms with low oxidation state; (iii) connectivity of the cluster units by common outer ligands. In spite of this, the anion valence violations are typical for most cluster compounds.

For example, in [Bu_4_N]_4_ [Re_6_S_8_Cl_6_], the inner and the outer ligands are presented by sulfur and chlorine, respectively. As a result, the valence violations (Figure 6a) are less pronounced than for pure chlorides or sulfides. In Nb_6_I_11,_ each outer ligand is common to two adjacent clusters, resulting in minimal valence violations for these ligands (Figure 6b). In Cs_4_Re_6_S_13_ different mechanisms of the valence compensation is working. The outer ligand S1 is common for two adjacent clusters, while the formal valence of the outer ligand S2 is not −2, but −1 (Figure 6c). In this compound, contributions of Cs^+^ cations in the BVSs of the inner and outer ligands are relatively close and do not effectively diminish the valence violations.

The clear tendency to reduce the valence violations, which appears in a large part of cluster compounds, seems to testify to a negative influence of the valence violations on material stability, but the absence of thermodynamic data on the stability of these compounds prevents the conclusion from being definitive.

### 3.3. Comparison with the Results of Quantum Chemistry Calculations

In this section, we try to analyze how the results of BVM application to compounds with metal–metal bonds agree with the data obtained by other methods. In Section 2, it was mentioned that we used a plethora of recent quantum chemistry data to confirm the validity of Equation (2) and to determine the BV parameters of the metal–metal bonds. Naturally, the BO values obtained by the BVM method for the TM–TM bonds agree well with quantum chemistry BO estimates. Moreover, Equation (2) is in good agreement with fundamentals of quantum chemistry, namely with exponential decay of the atomic and molecular wavefunctions (or orbitals), and thus may be qualitatively justified (see Ref. [13] for more details). Nevertheless, it is worth emphasizing that due to the high dispersion of quantum chemistry BOs (see Figure 1) obtained by using different recipes and wavefunction qualities, their exponential decay can be clearly established only in a wide range of interatomic distances and only for a sufficiently large variety of the BO data.

At first glance, it seems very simple to describe steric and electrostatic effects in cluster compounds by comparison of formal and effective BOs, regardless of the BO calculation method. However, with the exception of Corbett’s works mentioned above, these effects are known only due to the recent BVM application. The quantum chemistry studies rather compare the effective BOs obtained by different calculation methods, sometimes very complicated and time-consuming. In addition, they are focused solely on the metal–metal bonds, without analysis of respective valence violations for the metal–ligand bonds. Even in the rare cases that all the bonds were analyzed, the effects were not discussed.

As an example of such a rare case, Table 3 presents the results of the BV analysis based on the data by Baranovski and Korolkov for [Mo_6_S_8_(CN)_6_]^6−^. The bond lengths, R, and respective BOs calculated by quantum chemistry methods are taken from the original work [34]. Based on these data and geometry of the cluster unit (see the upper inset of Figure 3), we calculated the BVS separately for four Mo–Mo bonds in the Mo_6_ cluster, the BVS for four Mo–S bonds and a single Mo–C bond, as well as a total BVS for the Mo atom. A comparison between these BVSs and the formal values shows the same features that were discussed above: stretching of the Mo–Mo bonds and compression of the Mo–S bonds, while the total BVS of the Mo atom is close to the number of its valence electrons. The BVSs of the ligands decrease with the distance from the cluster center: they are high for the inner S ligands and low for the outer C ligands. It is worth noticing that the quantum chemistry work of Baranovski and Korolkov is not devoted to steric and electrostatic effects in cluster compounds. In spite of this, the results of this work are in excellent agreement with those obtained by the BVM application.

### 3.4. The BVM Application to the Electrode Materials with Metal–Metal Bonding

Compounds with metal–metal bonds often have interesting physical properties (optical, catalytic, thermoelectric, etc.). An excellent example is Chevrel phases, M_x_Mo_6_L_8_ (L = S, Se and Te, M is various cations). In addition to superconductive and thermoelectric applications [35,36,37,38], the sulfides and selenides were used as unique cathodes in Mg batteries [39]. The latter was suggested as promising analogs of well-known Li batteries, but a slow solid-state diffusion of divalent Mg^2+^ cations in most of the common hosts hampered their practical use [40]. In contrast, Chevrel phases allow for fast ionic transport of various multivalent cations [41]. One of the reasons for the high ionic conductivity is the presence of the Mo_6_ cluster, which can easily adopt up to four electrons upon cation insertion into Chevrel phase. This is in contrast to the individual TM atoms of the common hosts, which can simultaneously adopt only one electron. However, the BV analysis revealed additional bonding peculiarity in Chevrel phases, which may affect the ionic transport.

To illustrate this peculiarity, we chose Cu-containing Chevrel phases, Cu_x_Mo_6_S_8_, with accurately determined crystal structure [35], which is very close to that of Mg_x_Mo_6_S_8_ [42]. As can be seen in Figure 7, the BVSs of two crystallographically different sulfur atoms in Mo_6_S_8_ are completely different, in spite of the fact that both S1 and S2 atoms are inner ligands located at relatively close distances from the cluster center. The reason for the BVS distinction is their different bonding to the clusters (Figure 7a). S2 is connected only to one Mo_6_ cluster by three Mo–S bonds, while S1 has an additional bond to the adjacent cluster, for which it serves as the outer ligand. Thus, S1 has four Mo–S bonds with similar lengths. Figure 7b shows that Cu (or Mg) insertion into Mo_6_S_8_ is associated with the BVS growth for the S2 atom. As a result, the BVS distribution on the ligands becomes more uniform.

Such change of the ionic charges (BVSs) in the anion framework upon cation insertion in the electrode material is very unusual. It is commonly accepted that the charge of the inserted cations is compensated solely by the respective change in the oxidation state of transition metal. In contrast, in the Chevrel phases, the compensation also includes anions, thus facilitating the electron transfer in the crystal structure of the host. This mechanism of the charge distribution in Chevrel phases, first discovered by the BV analysis [8], was later confirmed by theoretical and experimental studies [43,44,45].

## 4. Conclusions

Recent application of the valence-length correlation proposed by Pauling about 70 years ago permits a simple, fast and zero-cost calculation of the effective BOs for the metal–metal bonds. Their comparison with the formal BOs, in combination with crystal structure rationalization, gives a unique possibility to describe steric and electrostatic effects, typical for the compounds with TM–TM interactions. These effects explain the bonding and structural peculiarities of cluster compounds. They seem to be crucial for the stability of these materials.

## Figures and Tables

**Figure 1 molecules-26-00304-f001:**
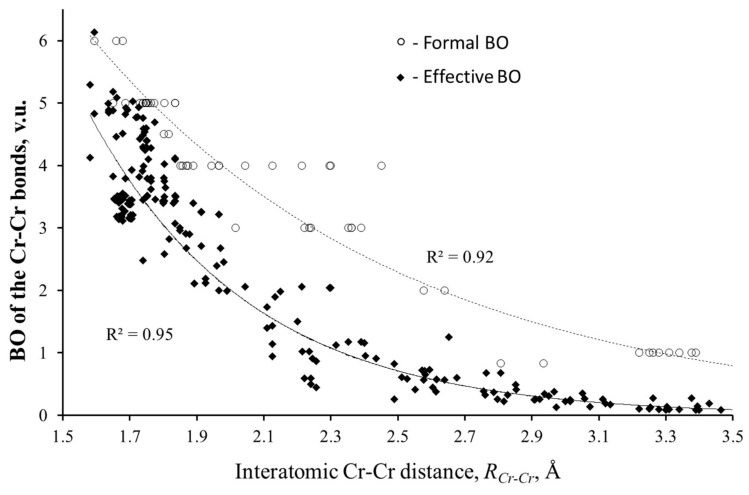
Formal and effective bond orders (BOs) of the Cr–Cr bonds vs. interatomic Cr–Cr distances.

**Figure 2 molecules-26-00304-f002:**
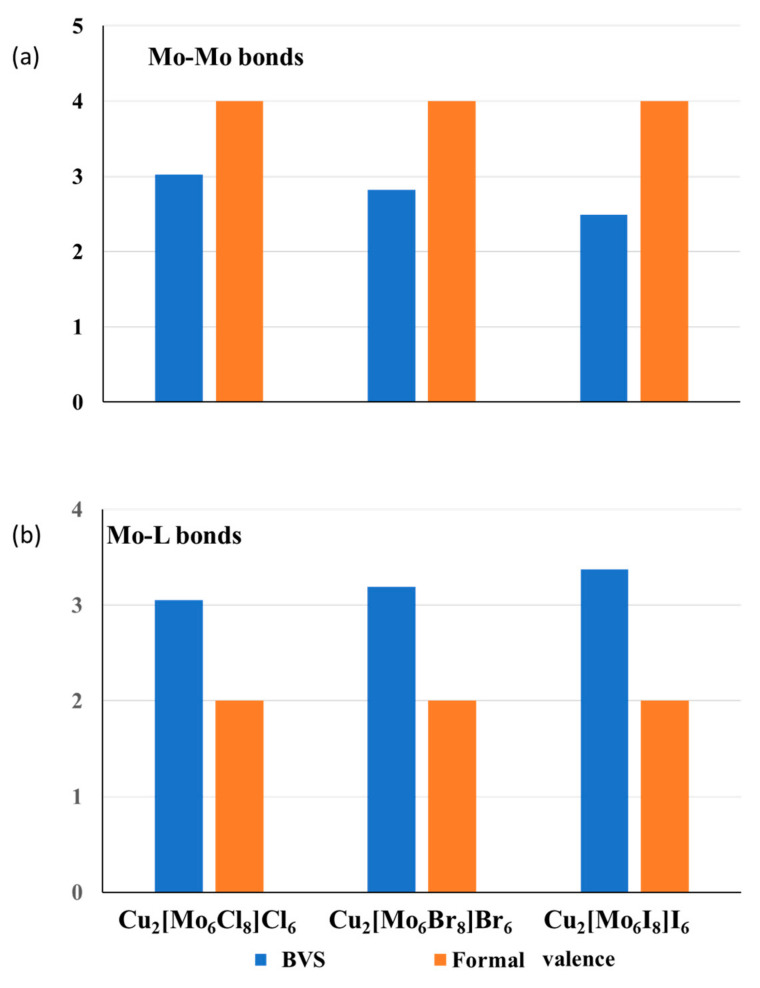
Effective anion charge (BVSs) and the formal numbers of valence electrons for the Mo–Mo (**a**) and Mo-L (**b**) bonds in Cu_2_[Mo_6_L^i^_8_]L^a^_6_ (L = Cl, Br and I) with octahedral Mo_6_ clusters coordinated by L^i^_8_-cube of the inner ligands and L^a^_6_-octahedron of the outer ligands.

**Figure 3 molecules-26-00304-f003:**
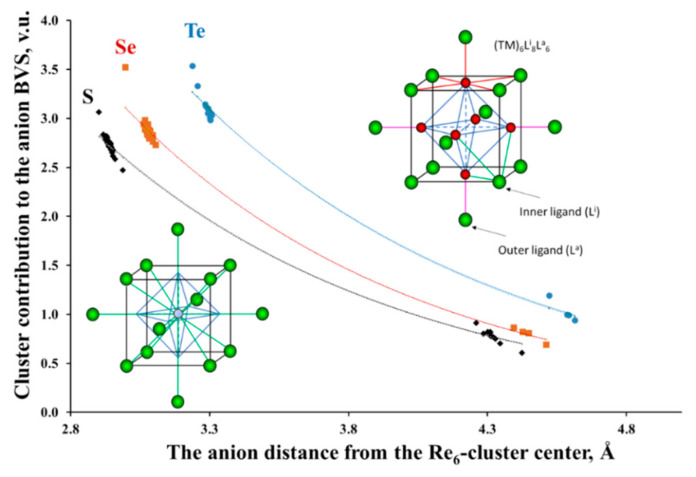
Re_6_ cluster contribution to the ligand valence as a function of their distance from the cluster center in cluster compounds with Re_6_L_8_-core (L = S, Se, Te). The upper inset shows the cluster unit Re_6_L^i^_8_L^a^_6_. The rhenium and chalcogen atoms are in red and green, respectively. Re-Re bonds in octahedral clusters are marked in blue. The Re-L bonds around Re, inner and outer ligands are in red, green and pink, respectively. In the low inset, the cluster is replaced by a single cation.

**Figure 4 molecules-26-00304-f004:**
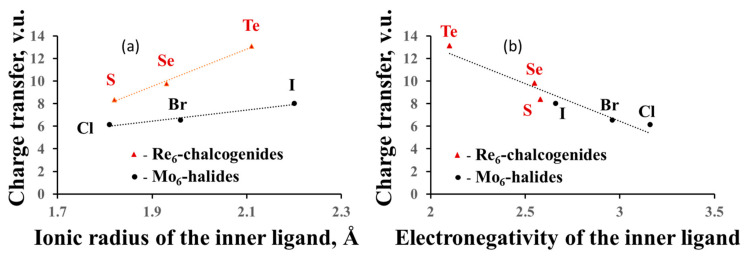
Charge transfer from the cluster to the ligands as a function of the size of the inner ligands (**a**) and the ligand electronegativity (**b**).

**Figure 5 molecules-26-00304-f005:**
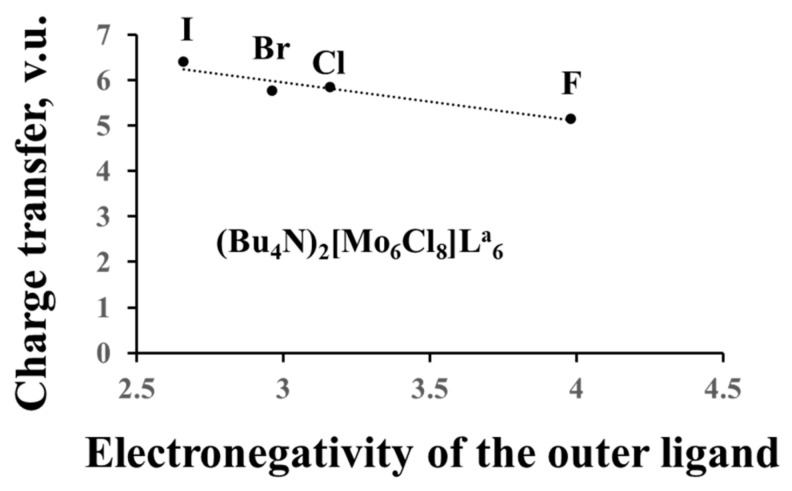
Charge transfer from the cluster to the ligands as a function of the electronegativity of the outer ligands.

**Figure 6 molecules-26-00304-f006:**
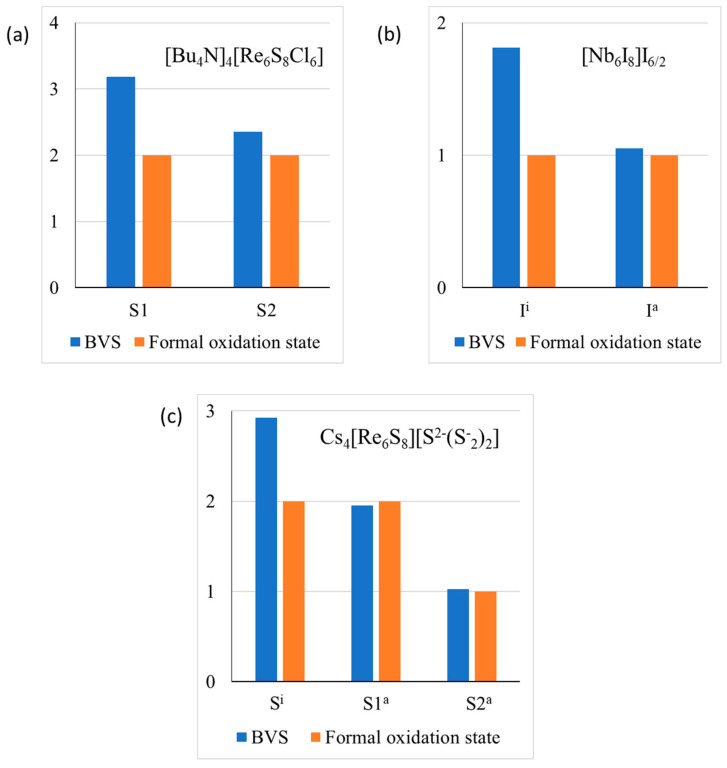
Valence violations for anions in [Bu_4_N]_4_ [Re_6_S_8_Cl_6_] (**a**), Nb_6_I_11_ (**b**) and Cs_4_Re_6_S_13_ (**c**). Here absolute values of the oxidation state are presented.

**Figure 7 molecules-26-00304-f007:**
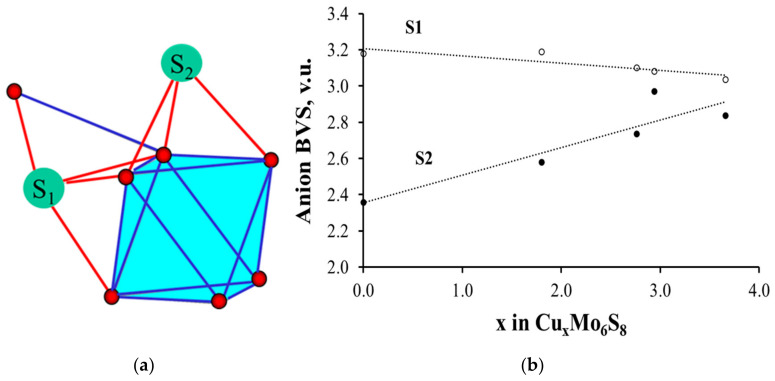
Different coordination of sulfur atoms in Chevrel phase (**a**) and the change of their BVSs in Cu_x_Mo_6_S_8_ (**b**).

**Table 1 molecules-26-00304-t001:** Comparison of the bond valence (BV) parameters for the metal–metal pairs used by Corbett [30] and Levi et al. [15].

Bond	Corbett, 1981			Levi et al, 2019
	*R_0_*, Å	*b*, Å	*R_0_*, Å	*b*, Å
Sc-Sc	2.921	0.26	2.695	0.28
Ti-Ti	2.638	0.26	2.505	0.47
V-V	2.464	0.26	2.435	0.43
Fe-Fe	2.367	0.26	2.26	0.44
Co-Co	2.323	0.26	2.11	0.44
Zr-Zr	2.918	0.26	2.89	0.43
Nb-Nb	2.708	0.26	2.64	0.395
Mo-Mo	2.619	0.26	2.51	0.34
W-W	2.635	0.26	2.535	0.29

**Table 2 molecules-26-00304-t002:** Bond valence parameters for the TM_6_-anion pairs (TM_6_ is a virtual cation located in the cluster center).

Bond	*R_0_*_TM6-L_, Å	*b*_TM6-L_, Å	Bond	*R_0_*_TM6-L_, Å	*b*_TM6-L_, Å
Nb_6_-F	2.96	1.36			
Nb_6_-Cl	3.66	1.07	W_6_-Cl	3.79	1.38
Nb_6_-Br	3.925	1.01	W_6_-Br	3.96	1.18
Nb_6_-I	4.2	1.12	W_6_-I	4.22	1.08
Mo_6_-Cl	3.75	1.25	Re_6_-S	4.03	1.09
Mo_6_-Br	3.905	1.14	Re_6_-Se	4.2	1.06
Mo_6_-I	4.17	1.07	Re_6_-Te	4.59	1.14

**Table 3 molecules-26-00304-t003:** BV analysis based on the data of Ref. [34].

Atom	Bonds	*R*, Å	*BO*, vu	*BVS*, vu	Formal Valence, vu	Valence Violations, vu
Mo	Mo-Mo	2.735	0.637	2.548	3.333	0.785
	Mo-L			3.381	2.667	−0.714
	All			5.929	6	0.071
S	Mo-S	2.539	0.731	2.193	−2	−0.193
C	Mo-C	2.305	0.457	0.457	−1	0.543

## Data Availability

Not applicable.

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
