# Peer review of "Metal–Metal Bond in the Light of Pauling’s Rules"

_molecules, 2021, doi:10.3390/molecules26020304_

Round 1
Reviewer 1 Report
I like this paper and can recommend publication as is. At the same time, the conclusions might be strengthened if the authors include into discussion also the high-pressure data that indicate that with increasing pressure metal-metal bonds often become longer, and this is compensated by the shortening of the metal - ligand bond. As examples one can see:
Casati, N., Macchi, P., & Sironi, A. (2009). Molecular crystals under high pressure: theoretical and experimental investigations of the solid–solid phase transitions in [Co2 (CO) 6 (XPh3) 2](X= P, As). Chemistry–A European Journal, 15(17), 4446-4457.
Resnati, G., Boldyreva, E., Bombicz, P., & Kawano, M. (2015). Supramolecular interactions in the solid state. IUCrJ, 2(6), 675-690.
Tidey, J. P., Wong, H. L., Schröder, M., & Blake, A. J. (2014). Structural chemistry of metal coordination complexes at high pressure. Coordination Chemistry Reviews, 277, 187-207.
Macchi, P., Casati, N., Evans, S. R., Gozzo, F., Simoncic, P., & Tiana, D. (2014). An “Off-axis” Mn–Mn bond in Mn 2 (CO) 10 at high pressure. Chemical communications, 50(85), 12824-12827.
Parois, P., Moggach, S. A., Lennie, A. R., Warren, J. E., Brechin, E. K., Murrie, M., & Parsons, S. (2010). The effect of pressure on the crystal structure of [Gd (PhCOO) 3 (DMF)] n to 3.7 GPa and the transition to a second phase at 5.0 GPa. Dalton Transactions, 39(30), 7004-7011.
Boldyreva, E. V. (2018). High Pressure Crystallography: Elucidating the Role of Intermolecular Interactions in Crystals of Organic and Coordination Compounds. In Understanding Intermolecular Interactions in the Solid State (pp. 32-97). Ed. by Deepak Chopra, Royal Society of Chemistry.
Author Response
We highly appreciate the Referee’s comment.
The bond valence analysis of the materials under high pressure is a very interesting issue, which needs a separate, profound study. We can expect two possible cases:
- Shortening of one kind of bonds due to the pressure is compensated by elongation of other kind of bonds, as suggested by the Referee.
- Pressure results in the valence violations and the changes in material stability. It seems that this case is typical for Chevrel phases (see Levi, E.; Aurbach, D. Chevrel Phases, MxMo6T8 (M = Metals, T = S, Se, Te) as a Structural Chameleon: Changes in the Rhombohedral Framework and Triclinic Distortion. Chem. Mater. 2010, 22, 3678–3692).
We have also to take into account that the bond valence parameters may be affected by pressure (see Brown, I. D.; Klages, P.; Skowron A. Influence of pressure on the lengths of chemical bonds. Acta Cryst. B 2003, B59, 439-448). This should complicate a proper bond valence analysis.

Reviewer 2 Report
This is an ok paper which tries to show the validity of the Pauling’s rules for compounds with metal-metal bonds. This paper complements in a way a recent perspective by the same authors recently published in PCCP devoted to the same topic. Nevertheless, I think that this works deserves publication in Molecules even if some parts, especially that devoted to Chevrel’s phases, are somewhat redundant with the PCCP paper. I just have a few comments that the authors should consider before publication proceeds.
- a) The authors are known to be experts in the use of Pauling’s rules to describe the bonding in inorganic compounds. They are not the only ones! In the past, J. K. Burdett and T. J. McLarnan (Amer. Mineral. (1984) 69, 601) tried to give an orbital interpretation of Pauling's rules. More recently, G. Hautier and colls. (Angew. Chem. Int. Ed. (2020) 59, 7569) have discussed the limited predictive power of the Pauling rules. This could be mentioned in the Introduction.
- b) Chevrel’s phases are amply discussed. However, most of the cited works come from the authors. At least some additional works – there are many – by others could also be cited.
- c) Fig. 7. A picture of the cluster with S1 and S2 atom labeling should accompany the plot.
- d) It is known that for the M6L14 clusters, electron counts from 20 to 24 are encountered. How is it interpreted by the BVS approach?
- e) As said earlier, the part devoted to the Cevrel phases and other M6L14 species has been amply discussed elsewhere. As other examples, the authors could have chosen the M6L18 compounds which are also widely encountered (see for instance P. Lemoine et al. Struct. Bond. 2019, 180, 143.
- f) Minor point. Pauling proposed his rules in 1929, over 90 years ago (not 60).
Author Response
Comment #1:
The authors are known to be experts in the use of Pauling’s rules to describe the bonding in inorganic compounds. They are not the only ones! In the past, J. K. Burdett and T. J. McLarnan (Amer. Mineral. (1984) 69, 601) tried to give an orbital interpretation of Pauling's rules. More recently, G. Hautier and colls. (Angew. Chem. Int. Ed. (2020) 59, 7569) have discussed the limited predictive power of the Pauling rules. This could be mentioned in the Introduction.
Answer on comment #1:
We highly appreciate the Referee’s comments.
In accordance with the Referee comment, the following sentence (marked in blue) and respective reference 6 were added to the Introduction (page 1, lines 43, 44):
"(Although some studies mark the limited predictive power of the Pauling Rules [6])."
Reference of J. K. Burdett and T. J. McLarnan (Amer. Mineral. (1984) 69, 601) was added to the part 2. Methods: Determination of the BV parameters for the metal-metal bonds (page 2, line 57) (see ref. 18).
Comment #2:
Chevrel’s phases are amply discussed. However, most of the cited works come from the authors. At least some additional works – there are many – by others could also be cited.
Answer on comment #2:
Four new references [35-38] for Chevrel phases were added on page 11, line 341.
Comment #3:
Fig. 7. A picture of the cluster with S1 and S2 atom labeling should accompany the plot.
Answer on comment #3:
The picture of the cluster with different coordination of S1 and S2 atoms was added to Fig. 7.
Comment #4:
It is known that for the M6L14 clusters, electron counts from 20 to 24 are encountered. How is it interpreted by the BVS approach?
Answer on comment #4:
According to Eq 1a, variations in the cluster electron count result in the changes in the formal bond valence (formal bond order) for the TM-TM bonds (the term S sTM-TM in Eq 1a). If all the electrons in the TM6-cluster are bonding, these changes are associated with respective changes in the oxidation state of the TM atom (the term S sTM-L in Eq 1a), while the total BVS of the TM atom remains the same. For example, for Mo6S8 (20-electron cluster) S sTM-TM = 3.333 (=20/6) and S sTM-L = 2.667. For Mg2Mo6S8 (24-electron cluster) S sTM-TM = 4 (=24/6) and S sTM-L = 2. The total BVS for the Mo atoms in the Mo6-cluster should be close to 6 (the number of the valence electrons of the Mo atom) for both compounds. Our calculations confirmed this.
To clarify this point, we added the following sentence (marked in blue) in the part 2 (Methods, page 2, lines 84-85):
"Details of the electron counting for various cluster compounds can be found in our previous works (see, for example, ref. 14)."
Comment #5:
As said earlier, the part devoted to the Cevrel phases and other M6L14 species has been amply discussed elsewhere. As other examples, the authors could have chosen the M6L18 compounds which are also widely encountered (see for instance P. Lemoine et al. Struct. Bond. 2019, 180, 143.
Answer on comment #5:
We chose Chevrel phases, because this is the best example to illustrate how, using the bond valence analysis, we can explain the unusually high ionic mobility in these materials. Thus, this part shows the practical importance of the bond valence analysis.
In our previous works the bond valence analysis was performed for numerous cluster compounds, including those with M6L18 cluster units proposed by the Referee (see, for example, the data for 12 Nb6-cluster compounds in ref. 14 (Levi, E.; Aurbach, D.; Gatti, C., Bond Order Conservation Principle and Peculiarities of the Metal-Metal Bonding. Inorg. Chem. 2018, 57, (24), 15550-15557).
Comment #6:
Minor point. Pauling proposed his rules in 1929, over 90 years ago (not 60).
Answer on comment #6:
Thank you for this correction. As was mentioned in the paper (the first sentence in the Introduction), the Pauling rules were proposed in 1929. However, the exponential correlation between the bond order and the bond length (expressed in the paper by Eq 2) was proposed later, in 1947. Thus, it was about 70 years ago, and we corrected this number in the abstract.

Reviewer 3 Report
I think the authors have done a heroic job in adding to our understanding of structural inorganic chemistry. At the risk of making a long and thorough paper even longer, why haven't the authors also discussed transition metal halide dimers such as Fe2Cl6 and Fe2Br4 (both studied by electron diffraction by M Hargitttai and her coworkers) and the ever growing list of homoatomic oligomers of the transition metals. Certainly, the last of species obeys Pauling's conservation rule, And speaking of this latter rule, wouldn't suggest that any bond (save homoatomic dimers) be destabilized or even nonexistent.
My only remaining question is why is Cotton is cited alone when his book is coauthored?
Author Response
Comment #1:
I think the authors have done a heroic job in adding to our understanding of structural inorganic chemistry. At the risk of making a long and thorough paper even longer, why haven't the authors also discussed transition metal halide dimers such as Fe2Cl6 and Fe2Br4 (both studied by electron diffraction by M Hargitttai and her coworkers) and the ever growing list of homoatomic oligomers of the transition metals. Certainly, the last of species obeys Pauling's conservation rule, And speaking of this latter rule, wouldn't suggest that any bond (save homoatomic dimers) be destabilized or even nonexistent.
Answer on comment #1:
We highly appreciate the kind comments of the Referee.
It seems that Fe2Cl6 and Fe2Br4 are not good examples for our study, because, according to the data by Hargittai et al. (Z. Naturforsch. 1980, 35a, 848-851; J. Chem. Phys. 1991, 94, 7278-7286) mentioned by the Referee, the Fe-Fe distance (more than 3.2 Å) in these compounds is too long for the effective metal-metal interactions. These species really obey the Pauling’s conservation rule. For example, for Fe2Cl6, the BVS of both Cl atoms with coordination 1 and 2 is close to the expected number (1 v.u.).
In the previous works we performed the bond valence analysis, which includes also the Fe2 dimers such as Fe2(DPhF)3 and Fe2(DPhF)4. For comparison, the Fe-Fe distance in these compounds is 2.2-2.5 Å.
Comment #2:
My only remaining question is why is Cotton is cited alone when his book is coauthored?
Answer on comment #2:
As it is clear from the reference, the book is really written by a group of co-authors. However, the text cited in our paper is written by Cotton (see the part 16.1 Structural Correlations in the book).
To avoid the misunderstanding, we changed the words “ his book” by “the book” (page 2, line 90).
